# Atypical Ductal Hyperplasia after Vacuum-Assisted Breast Biopsy: Can We Reduce the Upgrade to Breast Cancer to an Acceptable Rate?

**DOI:** 10.3390/diagnostics11061120

**Published:** 2021-06-19

**Authors:** Luca Nicosia, Antuono Latronico, Francesca Addante, Rossella De Santis, Anna Carla Bozzini, Marta Montesano, Samuele Frassoni, Vincenzo Bagnardi, Giovanni Mazzarol, Oriana Pala, Matteo Lazzeroni, Germana Lissidini, Mauro Giuseppe Mastropasqua, Enrico Cassano

**Affiliations:** 1Department of Breast Radiology, IEO European Institute of Oncology, IRCCS, 20141 Milan, Italy; luca.nicosia@ieo.it (L.N.); antuono.latronico@ieo.it (A.L.); anna.bozzini@ieo.it (A.C.B.); marta.montesano@ieo.it (M.M.); enrico.cassano@ieo.it (E.C.); 2Department of Emergency and Organ Transplantation, Section of Anatomic Pathology, School of Medicine, University “Aldo Moro”, 70124 Bari, Italy; francesca.addante1@gmail.com; 3Postgraduate School in Radiology, University of Milan, 20122 Milan, Italy; rossella.desantis@unimi.it; 4Department of Statistics and Quantitative Methods, University of Milan-Bicocca, 20126 Milan, Italy; samuele.frassoni@unimib.it (S.F.); vincenzo.bagnardi@unimib.it (V.B.); 5Division of Pathology and Laboratory Medicine, IEO European Institute of Oncology, IRCCS, 20141 Milan, Italy; giovanni.mazzarol@ieo.it (G.M.); oriana.pala@ieo.it (O.P.); 6Division of Cancer Prevention and Genetics, IEO European Institute of Oncology IRCCS, 20141 Milan, Italy; matteo.lazzeroni@ieo.it; 7Division of Breast Surgery, IEO European Institute of Oncology, IRCCS, 20141 Milan, Italy; germana.lissidini@ieo.it

**Keywords:** breast biopsy, BIRADS, atypical duct hyperplasia, breast surgery, breast cancer, upgrade to cancer, overtreatment

## Abstract

(1) Background: to evaluate which factors can reduce the upgrade rate of atypical ductal hyperplasia (ADH) to in situ or invasive carcinoma in patients who underwent vacuum-assisted breast biopsy (VABB) and subsequent surgical excision. (2) Methods: 2955 VABBs were reviewed; 141 patients with a diagnosis of ADH were selected for subsequent surgical excision. The association between patients’ characteristics and the upgrade rate to breast cancer was evaluated in both univariate and multivariate analyses. (3) Results: the upgrade rates to ductal carcinoma in situ (DCIS) and invasive carcinoma (IC) were, respectively, 29.1% and 7.8%. The pooled upgrade rate to DCIS or IC was statistically lower at univariate analysis, considering the following parameters: complete removal of the lesion (*p*-value < 0.001); BIRADS ≤ 4a (*p*-value < 0.001); size of the lesion ≤15 mm (*p*-value: 0.002); age of the patients <50 years (*p*-value: 0.035). (4) Conclusions: the overall upgrade rate of ADH to DCIS or IC is high and, as already known, surgery should be recommended. However, ADH cases should always be discussed in multidisciplinary meetings: some parameters appear to be related to a lower upgrade rate. Patients presenting these parameters could be strictly followed up to avoid overtreatment.

## 1. Introduction

Breast lesions classified as having uncertain malignant potential (B3) on biopsy cause management challenges. Diagnostic improvement in the identification of breast lesions, together with the introduction of population-based mammographic screening programs, has led to an increased rate of B3 diagnoses. Atypical ductal hyperplasia (ADH) is one of the most frequent lesions observed. Mammographically detectable microcalcifications are typically associated [1]. ADH is morphologically defined as an epithelial intraductal proliferation with cytological and architectural features similar to those of low-grade ductal carcinoma in situ (DCIS), but with partial involvement of ducts and/or limited extension. ADH can exhibit different growth patterns (cribriform, micropapillary or solid) reaching up to 2 mm and is found in approximatively 1–10% of breast biopsies [1,2,3,4]. (Figure 1).

A review of the available literature shows that ADH is often associated with a significantly higher risk of concomitant DCIS and/or invasive carcinoma (IC) diagnosed by subsequent surgical excisions [5,6,7,8,9]. Although debate on this subject still exists, the diagnosis of ADH is an indication for surgery [3,6,10,11,12,13,14].

According to the Second International Consensus Conference on breast lesions of uncertain malignant potential [15], with the exception of ADH, minimally invasive management of B3 lesions with vacuum-assisted breast biopsy (VABB) continues to be an appropriate alternative to surgery in most cases. Conversely, ADH surgical excision is still recommended.

Follow up, without surgical excision, is infrequent and justified only in certain circumstances after multidisciplinary discussion. This may be due to the high percentage of biopsy-proven ADH lesions that are upgraded after subsequent surgical excision. The reported percentage of upgrade to DCIS and/or IC at the surgical excision after percutaneous breast biopsy is extremely variable in the literature, with values reaching up to 85% [16,17,18,19].

Clinical management of these lesions is based primarily on the risk of identifying carcinoma (either DCIS and/or IC) in the excision specimens [16,17,18,19]. In general, excision is usually recommended for ADH.

The aim of this study is to evaluate which factors, especially radiological, can influence the upgrade rate of ADH to in situ or invasive carcinoma in a representative group of patients who underwent VABB and subsequent surgical excision.

The identification of factors associated with diagnostic underestimation can be of great help in selecting, after a multidisciplinary meeting, those patients in which follow-up may be recommended rather than surgical intervention, thus avoiding overtreatment.

To reach this goal, we examined surgical specimens of patients diagnosed with ADH to identify potential indicators for upgrading.

## 2. Materials and Methods

We analyzed 2955 VABB performed at European Institute of Oncology (IEO, Milan, Italy) between January 2000 and December 2019 under ultrasound or stereotactic guidance. Of them, 141 were diagnosed as pure ADH lesions. All patients underwent subsequent surgical excision.

Lesions were classified according to the Breast Imaging Reporting and Data System (BI-RADS) [20].

The histological results of the biopsies were classified as B1 to B5 lesions, according to the UK B-coding system [21].

We selected lesions identified on mammograms or ultrasound as BI-RADS ≥ 3.

Most cases (123/141) were identified with screening mammography and a stereotactic VABB was performed using an 11 or 8 Gauge (G) needle.

In a few cases (18/141) lesion was identified during breast ultrasound performed for prevention in patients with dense breasts. In these cases, an ultrasound-guided biopsy was performed with a 10 G needle.

All patients who underwent stereotactic breast biopsies had two projection mammograms before the procedure (Figure 2).

The number of cores obtained for each biopsy was extracted from the pathological reports.

Following an ADH diagnosis of the biopsy specimen, surgical excision was performed in all patients.

The surgically obtained breast tissue specimens were grossly sampled following institutional guidelines. As a rule, surgical samples of patients who had previously undergone VABB were always X-rayed before gross examination to identify residual microcalcifications. Thereafter, the whole abnormal area, including residual adjacent fibrotic tissue, was paraffin-embedded and histologic sections were prepared and microscopically evaluated.

Each biopsy was individually compared with the corresponding excision specimen. In case of stereotactic VABB, all patients underwent two mammography projections after the biopsy in order to radiographically assess the complete removal of microcalcifications following the procedure. Analogously, in ultrasound-guided biopsies, an ultrasound scan was performed after the biopsy to assess the complete removal of the lesion following the procedure.

We evaluated the upgrade rate to breast cancer defined as the finding of a DCIS or IC in the surgical specimen.

We investigated a potential correlation between patient’s age, lesion size, BIRADS, number of cores, complete macroscopic removal of the lesion, cases showing ADH only in cores bearing microcalcifications, and the chance of upgrade to DCIS or IC in the surgical specimen.

We also explored a possible correlation between these parameters and the absence of further lesions at the subsequent surgical excision, including those cases showing only benign findings in the excision specimen.

Finally, we evaluated patient follow-up to look for signs of recurrence defined as any patient developing histologically proven ipsilateral or contralateral breast lesion (classified as B3, B4 or B5) detected by periodic radiological examinations (performed after surgery for ADH).

### Statistical Analysis

Continuous data are reported as medians and interquartile ranges. Categorical data are reported as counts and percentages.

Fisher’s exact test was performed to evaluate the association between patients’ characteristics and the four different events (benign findings in the absence of ADH or further lesion at surgical excision, DCIS, IC and a combination of carcinoma in situ/invasive carcinoma).

A multivariate logistic regression model was performed to evaluate the association between a combined outcome (DCIS or IC) and the variables associated with the combined outcome in the univariate analysis.

Predicted probabilities of the combined outcome, according to the multivariate logistic regression model, were calculated.

The cumulative incidence of lesion curve functions was estimated using the Kaplan–Meier method. The log-rank test was used to assess differences between patients with or without upgrade to carcinoma in situ or invasive carcinoma. Univariable Cox proportional hazard regression models were used to assess the association between patients’ characteristics and risk of lesion.

A *p*-value ≤ 0.05 was considered statistically significant.

All analyses were performed with the statistical software SAS 9.4 (SAS Institute, Cary, NC, USA).

## 3. Results

Of the 2955 breast biopsies performed over a 20-year period, 141 ADH cases were identified (the clinicopathological features of the patients are summarized in Table 1).

Of these, 123 were diagnosed by stereotactic biopsy and identified by mammography, while 18 cases were diagnosed by ultrasound-guided biopsy. Of the 123 stereotactic breast biopsies, all lesions were identified with mammography and showed only microcalcifications. The remaining 18 cases of our ADH population were detected as a nodule by ultrasound.

The median age of patients was 51 (45–59) years, the median number of cores per biopsy was 10 (8–13). The median size of the lesion was 15 mm (10–20 mm).

Radiological diagnoses were 5 BIRADS 3 (3.5%); 53 BIRADS 4a (37.6%); 52 BIRADS 4b (36.9%); 29 BIRADS 4c (20.6%); 2 BIRADS 5 (1.4%). Overall, in 66/141 cases (47.8%) the lesion (identified by mammography or ultrasound) was macroscopically removed by VABB.

On excision, considering all the 141 patients undergoing surgery, 11 (7.8%) were upgraded to IC and 41 (29.1%) were upgraded to DCIS. In detail, 31 out of 141 (22%) cases were upgraded to low grade DCIS; 9 out of 141 (6.4%) cases were upgraded to intermediate grade DCIS and 1 case (0.7%) was upgraded to high grade DCIS. In 47/141 (33.3%) cases, the diagnosis of ADH was confirmed in the surgical specimen. Conversely, in 42/141 (29.8%) cases, ADH was not found in subsequent surgical specimens and only benign findings were observed. Apparently, in these cases the ADH focus had been completely removed with the VABB procedure.

The pooled upgrade rate to DCIS or IC was statistically lower (Table 2) at univariate analysis considering these parameters: the complete removal of the lesion (*p*-value < 0.001); BIRADS ≤ 4a (*p*-value < 0.001); size of the lesion ≤ 15 mm (*p*-value: 0.002); age of the patients < 50 years (*p*-value: 0.035).

The presence of ADH only in specimens with microcalcifications resulted in a statistically significant lower chance of upgrading the lesion to IC (*p*-value: 0.031).

Moreover, by using patients’ characteristics which were statistically significant at the univariate analysis as independent variables, multivariate analysis showed a correlation between the removal of the lesion, BIRADS and age, with the probability of ADH upgrading at the subsequent surgical excision (Table 3).

According to this multivariate logistic regression, the predicted probabilities of upgrading the lesion (to in situ or invasive carcinoma) at surgical excision were calculated (Table 4).

45/141 patients (31.9%) were lost to follow up and not included in the statistical analysis. We found no significant correlation between the loss of patients to follow up and the prognostic variables (Figure 3 and Figure 4).

Finally, we observed that 12/96 patients (12.5%) presented a breast lesion during the follow up period (we excluded in the analysis 45 patients lost to follow up). Specifically, we observed 4 cases of low grade DCIS, 1 case of intermediate grade DCIS, 1 case of high grade DCIS, 3 cases of atypical lobular hyperplasia and 3 cases of IC. The median time from surgery to the finding of breast lesion at follow up was 2.9 years. The overall median time of follow up was 7.6 years (4.3–11.1).

Interestingly, considering all the 12 patients developing a breast lesion during the follow up period, 6 of them had already been upgraded at initial surgical excision (5 cases of low grade DCIS and 1 case of IC).

Finally, we did not find any significant association between patients’ characteristics and incidence of breast lesions (univariate analysis) during follow up following surgery for ADH (Table 5).

## 4. Discussion

In recent decades the widespread use of image-guided biopsy devices has increased the diagnosis of ADH, with a current observed incidence ranging from 1% to 10% [22,23]. Consequently, clinical awareness of lesion characteristics and clinical significance have gained broader attention [23,24,25,26,27,28]; however, their management is still under debate. The Second International Consensus Conference on breast lesions of uncertain malignant potential states that surgery should be performed [15].

Amorphous microcalcifications are the most common imaging presentation of ADH [28,29,30]. Our findings confirm these data, with microcalcifications being the most common form of presentation in patients diagnosed with ADH: most biopsies were therefore performed under stereotactic guidance. While ADH has no specific ultrasonographic findings, in our study it frequently appeared as a nodule by ultrasonography.

ADH diagnosed by breast biopsy is associated with a high incidence of DCIS and IC at the subsequent surgical excision [7,10,30,31,32], as demonstrated by recent metanalyses [33,34] reporting an upgrade rate of 9% for invasive carcinoma.

Our study is in line with these results: on surgical excision, we observed an upgrade rate of 7.8% to IC and of 29.1% to DCIS. Most of the upgrade cases (22%) were represented by low grade DCIS: this is probably associated with the ADH definition and the monocentric type of this study assuming the intra-observer reproducibility.

This result is even more important considering that follow-up, rather than surgery, is now proposed even in low-grade DCIS [35].

Our efforts should be focused, primarily, on trying to reduce the percentage of upgrade to IC. In our study the percentage of the pooled upstaging (7.8%) to IC is still too high to justify follow-up as a choice rather than surgery: according to the Breast Imaging Reporting and Data System, follow-up would be justified by an upgrade rate lower than 2% to IC [20]. Many studies have tried to identify features that may reduce the upgrade rate to IC and guide the clinical approach based on histopathological and imaging characteristics [36,37,38]. However, the upgrade to IC remains non-negligible and, according to Second International Consensus Conference recommendations [15], “surveillance can be justified only in special situations after discussion at the multidisciplinary meeting”.

In our study, we found a statistically significant reduction of the upgrade rate considering the following parameters: age of the patients, lesion size, BIRADs and the complete macroscopical removal of the lesion by VABB, as already documented for the upgrading rate of DCIS to IC [39]. Moreover, the upgrade rate to IC is significantly reduced when ADH is found only in specimens showing microcalcifications.

Multivariate logistic regression analysis showed that, considering all these parameters, the pooled percentage of upgrade at the surgical excision is considerably reduced. Moreover, in our work we highlight the importance of follow up, even after surgery, due to the non-negligible possibility of finding recurrent breast lesions. The importance of this result, in our study, is unfortunately limited by the high number of patients lost to follow up.

## 5. Conclusions

The overall upgrade rate of ADH to DCIS or IC is high, and surgery should be recommended. However, ADH cases should always be discussed in a multidisciplinary meeting in order to identify which parameters could be valuable to establish a risk score useful to distinguish patients that could be offered a short-term follow-up rather than immediate surgery. On the other hand, these conclusions are constrained by the retrospective nature of the study which represents the main limitation, along with the small number of ADH cases and the significant number of patients lost to follow-up.

Future studies involving multiple centers with different practices are needed to better describe the natural history of ADH and avoid unnecessary surgical interventions. A hypothetical trial specifically designed for ADH may randomize patients to surgical excision vs active surveillance or active surveillance combined with endocrine treatment (e.g., low dose tamoxifen), based on the results already achieved by other studies [40,41].

## Figures and Tables

**Figure 1 diagnostics-11-01120-f001:**
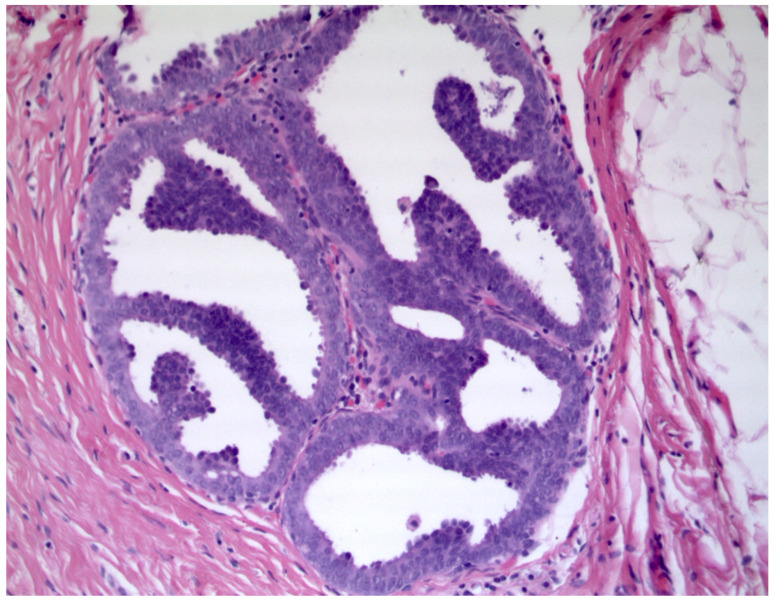
Atypical ductal hyperplasia with cribriform and micropapillary growth pattern (Hematoxylin & Eosin, 400×).

**Figure 2 diagnostics-11-01120-f002:**
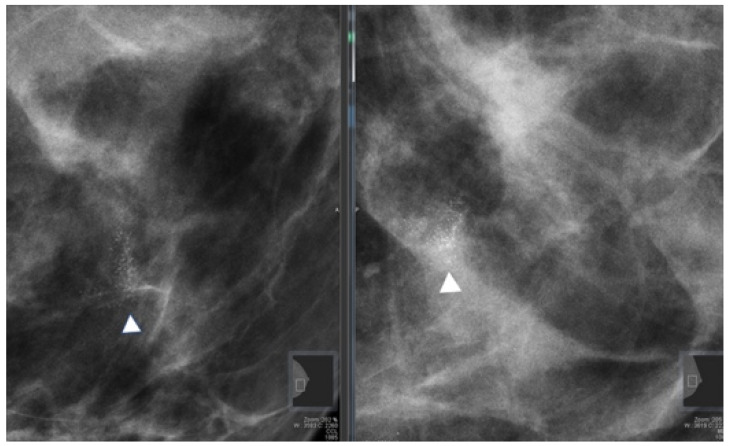
Full field digital mammography showing a cluster of microcalcifications (arrowhead) with a biopsy-proven histopathological result of atypical ductal hyperplasia.

**Figure 3 diagnostics-11-01120-f003:**
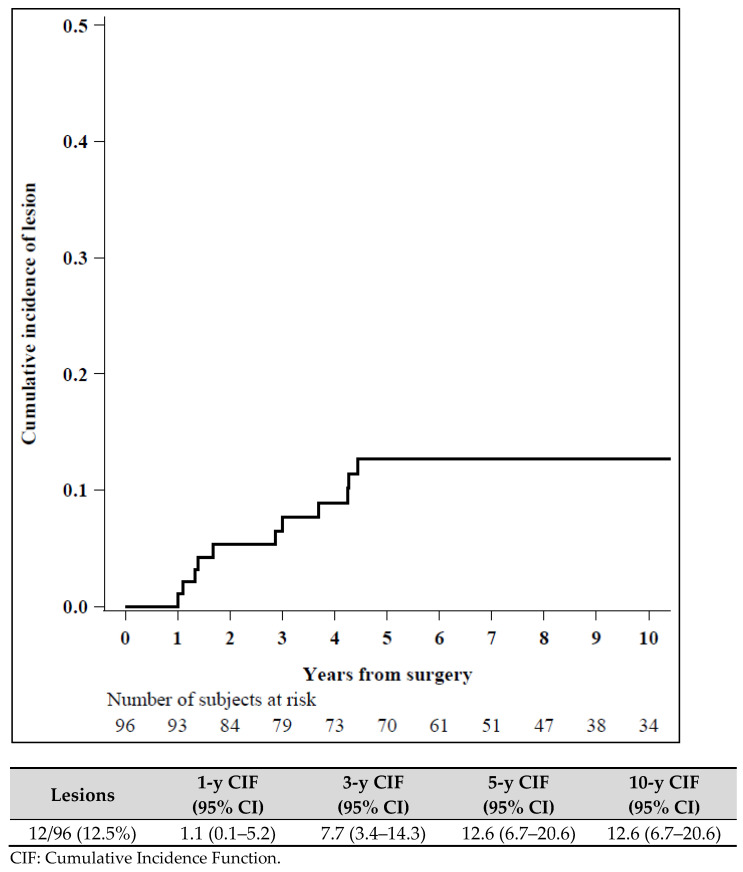
Cumulative incidence of lesion (N = 96).

**Figure 4 diagnostics-11-01120-f004:**
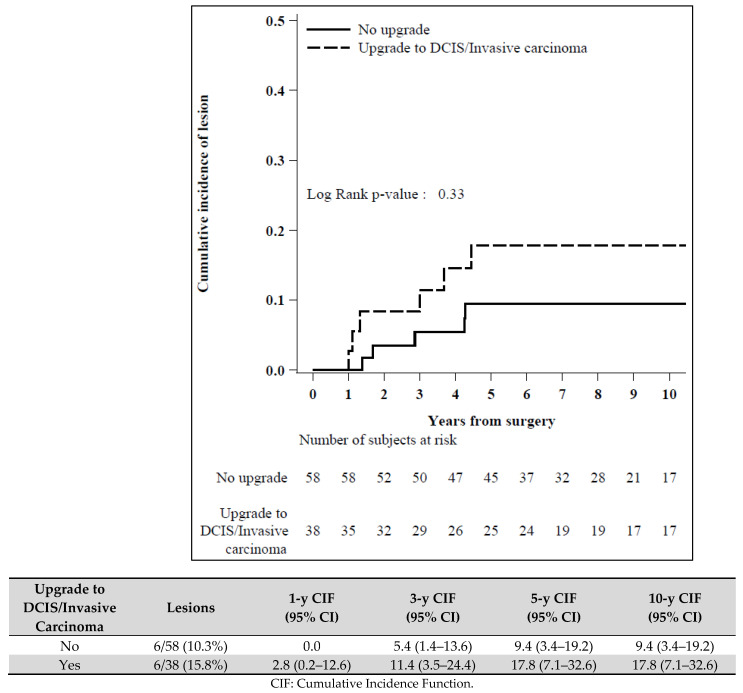
Cumulative incidence of lesion by upgrade to carcinoma in situ or invasive carcinoma (N = 96).

**Table 1 diagnostics-11-01120-t001:** Characteristics of patients (N = 141).

Variables	Overall (N = 141)
	N (%)
Age at biopsy (years)	
<40	2 (1.4)
40–49	58 (41.1)
50–59	50 (35.5)
60–69	22 (15.6)
70+	9 (6.4)
<50	60 (42.6)
50+	81 (57.4)
Median (IQR)	51 (45–59)
Days between biopsy and surgery	
≤30	24 (17.0)
31–60	41 (29.1)
61–90	30 (21.3)
>90	46 (32.6)
Median (IQR)	66 (41–112)
Size of the lesion (mm)	
≤15	89 (63.1)
>15	52 (36.9)
Median (IQR)	15 (10–20)
BIRADS classification of the lesion	
3	5 (3.5)
4a	53 (37.6)
4b	52 (36.9)
4c	29 (20.6)
5	2 (1.4)
3–4a	58 (41.1)
4b-4c-5	83 (58.9)
Number of cores	
<10	42 (29.8)
≥10	99 (70.2)
Median (IQR)	10 (8–13)
Imaging findings	
Microcalcifications	123 (87.2)
Nodule	17 (12.1)
Nodule with microcalcifications	1 (0.7)
Residual lesion at the biopsy *	
No	66 (47.8)
Yes	72 (52.2)
ADH only in cores with microcalcifications ^§^	
No	60 (51.3)
Yes	57 (48.7)

* 3 patients missing, ^§^ 24 patients missing.

**Table 2 diagnostics-11-01120-t002:** Association between patients’ characteristics and four outcomes (univariate analysis).

Variables	Confirmed Diagnosis of ADH	Lesion Removed	Carcinoma In Situ	Invasive Carcinoma	Event Considered:			
					Lesion Removed	Carcinoma In Situ	Invasive Carcinoma	Carcinoma In Situ or Invasive Carcinoma
	N (%)	N (%)	N (%)	N (%)	*p*-Value ^1^	*p*-Value ^1^	*p*-Value ^1^	*p*-Value ^1^
Overall	47 (33.3)	42 (29.8)	41 (29.1) ^2^	11 (7.8)				
Age at the biopsy (year)					0.19	0.086	0.21	0.035
<50	30 (50.0)	14 (23.3)	13 (21.7)	3 (5.0)				
50+	17 (21.0)	28 (34.6)	28 (34.6)	8 (9.9)				
Size of the lesion (mm)					0.13	0.002	0.29	0.002
≤15	34 (38.2)	31 (34.8)	18 (20.2)	6 (6.7)				
>15	13 (25.0)	11 (21.2)	23 (44.2)	5 (9.6)				
BIRADS classification of the lesion					0.001	<0.001	0.051	<0.001
3–4a	22 (37.9)	26 (44.8)	8 (13.8)	2 (3.4)				
4b–4c–5	25 (30.1)	16 (19.3)	33 (39.8)	9 (10.8)				
Number of cores					0.074	0.31	1.00	0.35
<10	16 (38.1)	8 (19.0)	15 (35.7)	3 (7.1)				
≥10	31 (31.3)	34 (34.3)	26 (26.3)	8 (8.1)				
Imaging findings					0.27	0.27	0.35	0.30
Microcalcifications	41 (33.3)	39 (31.7)	34 (27.6)	9 (7.3)				
Nodule/Nodule with microcalcifications	6 (33.3)	3 (16.7)	7 (38.9)	2 (11.1)				
Residual lesion at biopsy ^3^					<0.001	<0.001	0.20	<0.001
No	22 (33.3)	29 (43.9)	11 (16.7)	4 (6.1)				
Yes	24 (33.3)	11 (15.3)	30 (41.7)	7 (9.7)				
ADH only in cores with microcalcifications ^4^					0.42	1.00	0.031	0.44
No	20 (33.3)	16 (26.7)	16 (26.7)	8 (13.3)				
Yes	19 (33.3)	20 (35.1)	17 (29.8)	1 (1.8)				

^1^, Fisher’s exact test. ^2^, 31 low grade DCIS, 9 intermediate grade DCIS, 1 high grade DCIS. ^3^, 3 patients missing. ^4^, 24 patients missing.

**Table 3 diagnostics-11-01120-t003:** Multivariate analysis considering the upgrade from ADH to either DCIS or invasive disease as the outcome and patients’ characteristics resulting as statistically significant at univariate analysis (*p* < 0.05) as independent variables.

Variables	Upgrade to Carcinoma In Situ or Invasive Carcinoma/Tot (%)	Multivariate Analysis		
		OR	95% CI	*p*-Value
Overall	52/138 (37.7)			
Age at the biopsy (years)				
<50	16/58 (27.6)			
50+	36/80 (45.0)	2.53	1.11–5.80	0.028
Size of the lesion (mm)				
≤15	24/86 (27.9)			
>15	28/52 (53.8)	1.82	0.78–4.26	0.17
BIRADS classification of the lesion				
3–4a	10/57 (17.5)			
4b–4c–5	42/81 (51.9)	4.17	1.78–9.79	0.001
Residual lesion at the biopsy				
No	15/66 (22.7)			
Yes	37/72 (51.4)	3.02	1.27–7.22	0.013

Note 1. Only variables with *p* < 0.05 at univariate analysis were included in this analysis. Note 2. The 3 patients with missing value of “Residual lesion at the biopsy” were excluded from this analysis. OR: odds ratio; CI: confidence interval.

**Table 4 diagnostics-11-01120-t004:** Predicted probabilities of the upgrade from ADH to either DCIS or invasive disease, according to the multivariate logistic regression model.

Age at the Biopsy (Years)	Size of the Lesion (mm)	BIRADS Classification of the Lesion	Residual Lesion at the Biopsy	Probability (95% CI)
<50	≤15	3–4a	No	0.06 (0.02–0.15)
			Yes	0.15 (0.06–0.33)
		4b–4c–5	No	0.20 (0.09–0.38)
			Yes	0.42 (0.24–0.64)
	>15	3–4a	No	0.10 (0.03–0.27)
			Yes	0.24 (0.11–0.46)
		4b–4c–5	No	0.31 (0.13–0.58)
			Yes	0.57 (0.37–0.75)
50+	≤15	3–4a	No	0.13 (0.06–0.26)
			Yes	0.31 (0.14–0.55)
		4b–4c–5	No	0.38 (0.23–0.56)
			Yes	0.65 (0.44–0.81)
	>15	3–4a	No	0.21 (0.08–0.45)
			Yes	0.45 (0.24–0.68)
		4b–4c–5	No	0.53 (0.30–0.75)
			Yes	0.77 (0.61–0.88)

**Table 5 diagnostics-11-01120-t005:** Association between patients’ characteristics and incidence of lesion (univariate analysis) (N = 96).

Variable	N	Lesions	HR	95% CI	*p*-Value
Age at the biopsy (year)					
<50	46	5			
50+	50	7	1.25	0.40–3.97	0.70
Size of the lesion (mm)					
≤15	60	6			
>15	36	6	1.81	0.58–5.66	0.30
BIRADS classification of the lesion					
3–4a	44	4			
4b–4c–5	52	8	1.63	0.49–5.46	0.42
Number of cores					
<10	32	3			
≥10	64	9	1.23	0.33–4.60	0.75
Imaging findings					
Microcalcifications	82	11			
Nodule/Nodule with micro	14	1	0.60	0.08–4.63	0.62
Residual lesion at biopsy					
No	49	4			
Yes	47	8	2.34	0.70–7.78	0.17
ADH only in cores with micro					
No	37	4			
Yes	42	7	1.58	0.46–5.42	0.47
Missing	17	1			
Upgrade to DCIS/Invasive carcinoma					
No upgrade	58	6			
Upgrade	38	6	1.74	0.56–5.41	0.34

## Data Availability

Data sharing not applicable.

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
