# Peer review of "Atypical Ductal Hyperplasia after Vacuum-Assisted Breast Biopsy: Can We Reduce the Upgrade to Breast Cancer to an Acceptable Rate?"

_diagnostics, 2021, doi:10.3390/diagnostics11061120_

Round 1

Reviewer 1 Report

The manuscript “Atypical Ductal Hyperplasia after Vacuum assisted Breast Biopsy: can we reduce the upgrade to breast cancer to an acceptable rate?” by Nicosia et al. investigates patients who underwent vacuum-assisted breast biopsy for the rate of upgrade in post-surgical diagnosis from Atypical Ductal Hyperplasia to in situ or invasive carcinoma.

The paper presents a retrospective analysis of VABB cases, presumably performed at one hospital. The tables solely present basic descriptive statistics of the cases. Reasons for the diagnostic discrepancies are not elucidated. Presumably, all biopsies and surgical specimens were seen in the same pathology department. However, the manuscript shows only one mammogram and no pathology.

There could be value in this analysis. However, the sole reliance on basic descriptive statistics of case numbers in conjunction with a rather poor presentation of the writing render the manuscript unpublishable in the current form.

Specific points:

Almost every sentence in the results section has its own paragraph. This is not the way to write a manuscript.

The report contains a number of grammatical errors.

Abstract: replace “revised” with “reviewed”

Author Response

We thank the Reviewer for the suggestions.

The manuscript “Atypical Ductal Hyperplasia after Vacuum assisted Breast Biopsy: can we reduce the upgrade to breast cancer to an acceptable rate?” by Nicosia et al. investigates patients who underwent vacuum-assisted breast biopsy for the rate of upgrade in post-surgical diagnosis from Atypical Ductal Hyperplasia to in situ or invasive carcinoma.

The paper presents a retrospective analysis of VABB cases, presumably performed at one hospital.

Yes, the series is from a single Institution.

The tables solely present basic descriptive statistics of the cases.

We partially agree with this specific Reviewer’s comment. As a matter of fact, in Table 1 simple statistics describe baseline characteristics of patients included in the study, while in Tables 2, 3, 4 inferential statistics were reported, as described in “Statistical analysis” paragraph of the “Materials and Methods” section.

In detail:

- in Table 2, Fisher exact test was performed to evaluate the association between patients’ characteristics and the four different events (benign findings in the absence of ADH or further lesion at surgical excision; DCIS; IC; and a combination of carcinoma in situ/invasive carcinoma);

- in Table 3, a multivariate logistic regression model was performed to evaluate the association between a combined outcome (DCIS or IC) and the variables associated with the combined outcome in the univariate analysis;

- in Table 4, the predicted probabilities of the combined outcome were calculated according to the multivariate logistic regression model.

Reasons for the diagnostic discrepancies are not elucidated. Presumably, all biopsies and surgical specimens were seen in the same pathology department.

We agree with the Reviewer’s observation and actually, all biopsies and surgical specimens have been evaluated in the same pathology lab by the same pathologists as already abovementioned. We have added this piece of information in the Discussion section.

However, the manuscript shows only one mammogram and no pathology.

We thank the Reviewer’s suggestion. We have added a histological picture in the Introduction section.

There could be value in this analysis. However, the sole reliance on basic descriptive statistics of case numbers in conjunction with a rather poor presentation of the writing render the manuscript unpublishable in the current form.

We hope that, thanks to the interesting Reviewer’s comments, our revision has made the paper worthy of publication.

Specific points:

Almost every sentence in the results section has its own paragraph. This is not the way to write a manuscript.

Actually, we agree with the Reviewer’s comment and according to that, we have reduced the number of paragraphs.

The report contains a number of grammatical errors.

As already written, we send the manuscript to a Mother Tongue writer in order to render the manuscript suitable for publication.

Abstract: replace “revised” with “reviewed”

We have replaced the verb, according to the appropriate Reviewer’s suggestion.

Reviewer 2 Report

It is a very interesting paper supported by significative numbers, although ADH has a known underestimation problem

It is a very interesting paper with a very strong research group.
Their experience about B3 and in detail about ADH lesions. 
In clinical practice, ADH substime is a very dilemma of breast radiologist and this paper could be very useful for radiologists. 
Vabb results and surgery comparison should help in patient management. 
English is good, statistic is strong and conclusion are in agreement with results. 

Author Response

It is a very interesting paper supported by significative numbers, although ADH has a known underestimation problem

It is a very interesting paper with a very strong research group.

Their experience about B3 and in detail about ADH lesions.

In clinical practice, ADH substime is a very dilemma of breast radiologist and this paper could be very useful for radiologists.

Vabb results and surgery comparison should help in patient management.

English is good, statistic is strong and conclusion are in agreement with results.

We thank the Reviewer for the positive and encouraging comments.

Reviewer 3 Report

The authors present a very interesting analysis of patients diagnosed with ADH in order. to verify the likelihood of upgrading their diagnoses to definite malignancies.

They are to be congratulated on the study design and the data analysis performed.

The exact length of time considered in the study should be clarified (in the methods section it is stated that it is between January 2000 and December 2019, but in the results section it refers to a 10-years period)

The discussion section should be improved, in particular the limitations of this study should be clarified and discussed.

Author Response

The authors present a very interesting analysis of patients diagnosed with ADH in order. to verify the likelihood of upgrading their diagnoses to definite malignancies.

They are to be congratulated on the study design and the data analysis performed.

The exact length of time considered in the study should be clarified (in the methods section it is stated that it is between January 2000 and December 2019, but in the results section it refers to a 10-years period).

We thank the Reviewer for the encouraging comments and we apologize for the typo about the period of the study in the Result section. We have corrected the number of years.

The discussion section should be improved, in particular the limitations of this study should be clarified and discussed.

We have clarified the monocentric nature of the study in the Discussion section and we have commented about the limitations of the study in the Conclusions section.

Reviewer 4 Report

Thank you for asking me to review this manuscript which evaluates the probability of upgrade from ADH to DCIS or invasive disease in a cohort of patients undergoing vacuum assisted biopsy. It is a relatively small series of 141 patients from a single centre.

In univariate analysis (Table 3) the authors suggest that age >50, lesion size, BI-RADS classification and residual lesion are predictors of upgrade to either DCIS or invasive disease.

Were all cases detected on screening mammography? Was there a retrospective review of the imaging to classify lesion size and BI-RADS classification or was this recorded propsectively?

Why did some cases have US guided VABB?

Were marker clips placed at VABB to ensure that surgical excision of the correct area was carried out, especially in cases where the lesion was completely excised at VABB? This may affect upgrade rates if the correct area was not excised (and this can be difficult to identify following VABB).

Table 4 presents a multivariate analysis, presented by age, lesion size and BI-RADS category, but the numbers in each category must be very small - can patient numbers be included in this table please? I find this table quite confusing - is what is being reported in the 'Probability' column the probability of upgrade to either DCIS or invasive disease? As noted by the authors the important consideration is upgrade to invasion rather than DCIS (as this is often low grade and could be considered not to require further excision). Could the authors consider presenting these probabilities separately?

Given the historical nature of the series, can the authors present any follow up data? The paper would be hugely strengthened if they could show rates of ipsilateral invasive or in situ disease events in this cohort.

Based on their findings, in their conclusions can the authors make any recommendations about patients who do not require further excision after VABB due to a very low risk of upgrade? This would be the most important conclusion from a clinician and patient perspective and I would like to see some stronger conclusions being drawn based on this data.

Author Response

Thank you for asking me to review this manuscript which evaluates the probability of upgrade from ADH to DCIS or invasive disease in a cohort of patients undergoing vacuum assisted biopsy. It is a relatively small series of 141 patients from a single centre.

In univariate analysis (Table 3) the authors suggest that age >50, lesion size, BI-RADS classification and residual lesion are predictors of upgrade to either DCIS or invasive disease.

Were all cases detected on screening mammography? Was there a retrospective review of the imaging to classify lesion size and BI-RADS classification or was this recorded propsectively?

Why did some cases have US guided VABB?

We thank the Reviewer for giving us the opportunity of adding this piece of information.

Actually, most cases (123/141 e.g. 87.2%) were identified with screening mammography while in a minority of our population (18/141), the lesion was identified during breast ultrasound performed for prevention in patients with dense breasts. BIRADS were prospectively recorded.

We added these data in “Materials and Methods” section.

Were marker clips placed at VABB to ensure that surgical excision of the correct area was carried out, especially in cases where the lesion was completely excised at VABB? This may affect upgrade rates if the correct area was not excised (and this can be difficult to identify following VABB).

We thank the Reviewer for the interesting observation and question.

In our series, a clip was always placed after the VABB with complete removal of the lesion. In patients where the lesion was not completely removed by biopsy, the clip was not positioned: however, a pre-operative radiological mapping, using permanent marker, was always performed the day before surgery in order to make the surgeon able to identify the exact site of the lesion.

Table 4 presents a multivariate analysis, presented by age, lesion size and BI-RADS category, but the numbers in each category must be very small - can patient numbers be included in this table please? I find this table quite confusing - is what is being reported in the 'Probability' column the probability of upgrade to either DCIS or invasive disease? As noted by the authors the important consideration is upgrade to invasion rather than DCIS (as this is often low grade and could be considered not to require further excision). Could the authors consider presenting these probabilities separately?

The results of multivariate analysis, along with number of events and patients at risk for each variable included in the model, is presented in Table 3. Table 4, on the other hand, shows the predicted absolute risks of event for each unique combination of the four binary variables considered in the multivariate model presented in table 3 (the number of combinations is 24=16). This is an alternative way to present the results of the same multivariate model, and, since absolute risks are more easily interpretable than odds ratio, we think that this piece of information could help clinicians to better understand the results of our model.

The events considered in the model were, as detailed in both the titles of tables 3 and 4, “Carcinoma in situ or invasive carcinoma”. Since the Reviewer found these definition confusing, we rephrase it in the new version of the paper, where the terms “upgrade from ADH to either DCIS or invasive disease” were used instead.

We agree with the Reviewer that a model considering, as the outcome of interest, the upgrade limited to invasion rather than DCIS would have been more clinically relevant. However, the upgrade to invasive breast cancer was observed in only 11 patients in our sample (as presented in table 2): with such a small number of events, the results of a multivariable model are not reliable.

Given the historical nature of the series, can the authors present any follow up data? The paper would be hugely strengthened if they could show rates of ipsilateral invasive or in situ disease events in this cohort.

We thank the Reviewer for the comment and followed the suggestion to add part of follow up data and additional statistical analyses in the text.

Based on their findings, in their conclusions can the authors make any recommendations about patients who do not require further excision after VABB due to a very low risk of upgrade? This would be the most important conclusion from a clinician and patient perspective and I would like to see some stronger conclusions being drawn based on this data.

We thank the Reviewer for the fascinating comment. We added this recommendation in the “Conclusions” section with its proper refs:

“Future studies involving multiple centers with different practices are needed to better describe the natural history of ADH and avoid patients from unnecessary surgical interventions. A hypothetical trial specifically designed for ADH may randomize patients to Surgical Excision vs Active Surveillance or Active Surveillance combined with endocrine treatment (e.g. low dose tamoxifen), based on the results already achieved by other studies".

Round 2

Reviewer 1 Report

The manuscript has undergone substantial language editing. However, the concerns expressed in the original review were much more extensive. Contrasting with the recommendation for major revision, the content of the paper has only undergone minor changes. Therefore, I regret that I cannot recommend further consideration of the report.

Author Response

We thank the Reviewer for the comments.

We have changed the text according to other Reviewers searching for satisfying every comments.

Reviewer 4 Report

Thank you taking into consideration my previous comments, which have largely been addressed in this revised version of the manuscript.

Regarding the follow-up, I have two minor comments. What was the median follow-up time overall? It's stated that 8.5% developed a breast lesion during follow up with a median time to developing a lesion of 2.9 years - however it would be good to know what the overall length of follow up was. Also, almost one third of patients were lost to follow up and this needs to be mentioned as a limitation in the discussion section where the authors are highlight the importance of follow up - the actual rate of developing subsequent breast lesions may have been higher than the reported 8.5% due to loss to follow up.

Finally, for their proposed trial in the Conclusion section - do the authors consider there to be a role for VAE as a therapeutic approach as an alternative to surgery?

Author Response

Thank you taking into consideration my previous comments, which have largely been addressed in this revised version of the manuscript.

Regarding the follow-up, I have two minor comments. What was the median follow-up time overall? It's stated that 8.5% developed a breast lesion during follow up with a median time to developing a lesion of 2.9 years - however it would be good to know what the overall length of follow up was.

We thank the Reviewer for the comment.

“The median time of follow up was 7.6 years (4.3-11.1).”

This phrase was already reported in the text in the Materials and methods section; actually, we moved the sentence in the Results section.

Also, almost one third of patients were lost to follow up and this needs to be mentioned as a limitation in the discussion section where the authors are highlight the importance of follow up  the actual rate of developing subsequent breast lesions may have been higher than the reported 8.5% due to loss to follow up.

We thank the Reviewer for the valuable observation.

We changed the text according to the suggestion and modified the discussion section mentioning this limiting point.

Additionally, taking into consideration the Reviewer’s advice, we changed the Results section according to patients lost to follow-up.

Finally, for their proposed trial in the Conclusion section - do the authors consider there to be a role for VAE as a therapeutic approach as an alternative to surgery?

The main purpose of our study is to select some parameters, mostly radiological (eg complete removal of the lesion on biopsy, diameter of the lesion etc.) which are associated with a lower risk of upgrade to subsequent surgery.

This could be very useful for prospective trials in which a “watch-and-wait” approach is proposed instead of surgery for patients with low risk of upgrade.

The main purpose of our study is therefore not to propose VABB as an alternative to surgery, but to safely avoid surgery in selected patients.

Interestingly, however, in our study we found that complete removal of the lesion with VABB is associated with a lower risk of upgrade to subsequent surgery.
